# Comparison of Various Drought Resistance Traits in Soybean (*Glycine max* L.) Based on Image Analysis for Precision Agriculture

**DOI:** 10.3390/plants12122331

**Published:** 2023-06-15

**Authors:** JaeYoung Kim, Chaewon Lee, JiEun Park, Nyunhee Kim, Song-Lim Kim, JeongHo Baek, Yong-Suk Chung, Kyunghwan Kim

**Affiliations:** 1Gene Engineering Division, Department of Agricultural Biotechnology, National Institute of Agricultural Science, Jeonju-si 55365, Republic of Korea; jaeyoung7798@jbnu.ac.kr (J.K.); knh702@korea.kr (N.K.); greenksl5405@korea.kr (S.-L.K.); firstleon@korea.kr (J.B.); 2Crop Cultivation & Environment Research Division, National Institute of Crop Science, Suwon-si 16613, Republic of Korea; wowlek44@korea.kr; 3Department of Plant Resources and Environment, Jeju National University, Jeju-si 63243, Republic of Korea; ag0526@jejunu.ac.kr

**Keywords:** abiotic stress response, digital image analysis, image processing, phenotyping platform system, RGB phenotyping

## Abstract

Drought is being annually exacerbated by recent global warming, leading to crucial damage of crop growth and final yields. Soybean, one of the most consumed crops worldwide, has also been affected in the process. The development of a resistant cultivar is required to solve this problem, which is considered the most efficient method for crop producers. To accelerate breeding cycles, genetic engineering and high-throughput phenotyping technologies have replaced conventional breeding methods. However, the current novel phenotyping method still needs to be optimized by species and varieties. Therefore, we aimed to assess the most appropriate and effective phenotypes for evaluating drought stress by applying a high-throughput image-based method on the nested association mapping (NAM) population of soybeans. The acquired image-based traits from the phenotyping platform were divided into three large categories—area, boundary, and color—and demonstrated an aspect for each characteristic. Analysis on categorized traits interpreted stress responses in morphological and physiological changes. The evaluation of drought stress regardless of varieties was possible by combining various image-based traits. We might suggest that a combination of image-based traits obtained using computer vision can be more efficient than using only one trait for the precision agriculture.

## 1. Introduction

Rapid climate changes and global warming have led to the emergence of world food security as a rising issue in recent agricultural studies and industries [1]. Severe drought can critically damage the plant life cycle and its yield through an environment of heavy water deficit at the vegetative and reproductive stages [2,3]. The annual increase in drought severity interferes with agricultural sustainability through crucial physical, physiological, and biochemical damages [4]. Drought stress in plants is indicated by low leaf water potential and turgor pressure, increased stomatal closure, decreased cell activities, photosynthesis, nutrient uptake, and translocation in physiological disorders [5]. Various methods were suggested to overcome drought stress. Breeding drought-resistant cultivars is considered one of the most efficient methods for coping with frequent and severe water deficit conditions [6]. However, conventional plant breeding used over the last few decades in genetic/phenotypic selections is a time-consuming system. The advance in modern genetic technology has lessened the time cost and breeding cycle by developing genome editing techniques such as ZFNs (zinc-finger nucleases), TALENs (transcription activator-like endonucleases), and CRISPR/Cas9 (CRISPR-associated Cas9 endonucleases), while phenotypic analysis and selection are considered conventional methods [7].

Time consumption is not the only issue in breeding drought-resistant cultivars. The other issues include high labor intensity, destructive measurements, and inaccurate evaluation. The recent emergence of high-throughput phenotyping methods has led to the feasible, rapid, accurate, nondestructive acquisition of massive data and analysis, replacing manual measurement with an automated process using spectral sensors, computer vision, and digital image processing [8]. The application of digital image data in the phenotyping procedure, with the appropriate use of spectral sensors, promises the efficient analysis of targeted phenotypes [9]. Studies have reported the evaluation of the status of plant health in water and nutrition deficit conditions using the leaf area index [10], biomass, height [11], growth monitoring [12], and other traits. This was achieved with lower labor intensity and improved accuracy by applying the RGB sensors, which allow images of visible band wavelengths [13]. However, despite the studies conducted so far, image analysis-based traits demonstrate different patterns for each cultivar and species. A more in-depth study of image-based phenotype for each crop species and cultivar is, therefore, required.

Soybean (*Glycine max* L.) is one of the most consumed crops worldwide and is used as a principal food source and in various industries. Soybean is prone to drought during the cultivation season [14] and requires sufficient water from the beginning of growth to the reproductive period. Soybeans have several easy-to-observe phenotypes demonstrating an immediate response to an environment of water shortage, such as the shape and color of the leaves, leaf production rates, and the curvature of the plant. Studies have demonstrated that drought stress during reproductive stages negatively affects the vegetative status, the leaf water content, the biomass of soybean reproductive organs, the antioxidant capacity, the seed weight, the number of seeds per pod, and protein/nutritional compositions, all finally resulting in yield loss [15]. The effects of water deficit stress on developmental stages are known to decrease the soybean leaf area, the stem internode elongation, and the plant height [16], which have relatively fewer studies conducted than the studies on reproductive stage responses. However, in the case of high-throughput phenotyping methods, the plant vigor in vegetative stages and immediate responses may allow for the evaluation of effective stress due to its image-based analysis system. This feature is expected to contribute to the fastening of breeding cycles in developing resistant cultivars by enabling selection in the early vegetative stages.

Several studies on soybean abiotic stress responses have suggested the application of computer vision techniques for their evaluation [17,18,19]. This study focused on investigating the effective and optimized image-based phenotypes for monitoring drought tolerant traits, drought stress evaluation, and recovery rate assessments on different early vegetative stages of 28 varieties of soybean nested association mapping (NAM) population parents. Quantitative drought traits were generated through an image-based phenotyping method using the digitized measurement and analysis system. The overall and individual image-based traits were analyzed using statistics such as principal component analysis (PCA) and t-distributed stochastic neighbor embedding (t-SNE) algorithms.

The differentiation of phenotypic traits among different species and varieties of crops creates a bottleneck in the application of high-throughput phenotyping methods. Moreover, we were able to identify individuals those had experienced drought stress even after the recovery period. This result may be helpful to researchers who require image-based trait selection for precision agriculture related to drought stress evaluation, since the individuals were not easily recognizable through a single image-based trait or with the naked eye.

## 2. Results

### 2.1. Validation with Actual Measured Data

The actual data obtained from soybeans were analyzed for validating drought treatment effects and image-based data. Height, number of main stem nodes, total nodes, and number of produced pods were measured by actual manors. These traits are well known to develop during vegetative growth and influence the formation of pods, as yield indicators [20].

The Kruskal–Wallis analysis of variance indicated that the height and number of nodes of drought treated and untreated groups at all vegetative stages were not influenced (Table 1). However, the late divergence of V4 at 7 days after the drought treatment appears to be the result of sufficient time provided for their vegetative growth. In addition, although soybeans that were drought treated at the V2 and V3 stages ended up with no difference in the number of node formations, V4 soybeans showed a difference between both treatment groups at the end of recovery. The number of produced pods showed no differences at V3 and V4. During the second vegetative stage set, a delay in measuring plant height occurred from mis-scheduling. As a result, we validated their conditions by counting nodes instead, while height was eventually measured at two other stages (V3 and V4).

The Pearson correlation analysis was primarily conducted twice between the image-based and actual measured traits. First, the number of main stem nodes, total nodes, and area (Figure 1), and height and horizontal/vertical image-based areas as second (Figure 2). Both figures showed that all traits have some medium to strong levels of positive correlations with each other. Moreover, image-based traits and actual measured traits at V4 had the least correlation coefficient levels among the three sets (Figure 1c and Figure 2c).

### 2.2. Area Related Image-Based Traits

The image-based traits, such as area, convex hull area, and object sum area, demonstrated similar patterns in both vertical and horizontal angled images at the third vegetative stage (V3). They also showed no differences among the varieties at V3 (Table 2). The vertical angled caliper length at the second vegetative stage (V2) had no difference between the unstressed and drought-treated. The min area rec-tangle area showed no differences at the vertical images of V3, and the horizontal images of the fourth vegetative stage (V4).

PCA in both horizontal and vertical images was conducted to determine their differences and evaluation of each area trait. All area categorized traits (area, caliper length, convex hull area, min area rectangle area, and object sum area) were performed. The PCAs from horizontal images were not used due to their inaccurate results. The vertical image traits were separated by drought treatment with some overlapping clustering zones between the drought treated and untreated individuals (Figure 3).

The overlapping seems to have occurred from fewer or no differences in the beginning of drought treatment (Figure 4). Additionally, Common, NAM01, NAM05, NAM06, and NAM11 showed particularly fewer or no differences at the beginning of drought treatment among the 24 varieties in horizontal angled images (Figure 4a) than the vertical images (Figure 4b). We presume that this might be the reason for the large overlapping in PCA plots for horizontal images. The missing data of V3 before drought treatment, caused due to system errors, were excluded (Figure 3a).

### 2.3. Boundary Related Image-Based Traits

The image-based traits related to boundary (boundary point roundness, circumference, convex hull circumference, min enclosing circle diameter, and roundness) demonstrated different appearances depending on the angle of image capture. From the Kruskal–Wallis analysis, vertical boundary point roundness was the only trait that had no differences in drought-treated and unstressed individuals (Table 3). The others showed weak effects from the varieties mostly at the vertical images. The results of PCA in Figure 5 showed that the unstressed and drought-treated individuals were closely clustered together at all time periods, as indicated in Table 3. The missing data of V3 before drought treatment, caused due to system errors, were excluded (Figure 5a).

### 2.4. Color Related Image-Based Traits

Analyzing the RGB values is certainly a strong indicator of drought. However, the evaluation of RGB values requires a large amount of pixel data in all image files to be analyzed each. The processing of all images enlarges the time and hardware requirements; therefore, we used the mean variance of RGB values for a quick and efficient evaluation process.

The mean color variance of each red, green, and blue bands (mean color red, green, blue variance) assessed the changes in leaf status of the drought treated individuals during the treatment. The ‘Mean color red variance’ is the variance of R band values, related to brightness. Based on 126, the closer to 0 is black, and the closer to 255 is white. The ‘Mean color green variance’ is a parameter that indicates whether the color is biased toward green or red calculated from the green band. Based on 126, when analyzing the result value, the color closer to 0 is green, and the closer to 255, the color is biased toward red. The ‘Mean color blue variance’ is the blue band values which indicate whether the color is biased toward blue or yellow. Based on 126, when analyzing the result value, the closer to 0 is the blue color, and the closer to 255, the color is biased toward the yellow color. Thus, these parameters inform simplified fluctuation of color values occurring in the drought stressed individuals (Figure 6).

The red and green band values followed similar patterns in both unstressed and drought-treated groups. The harsh changes in drought-treated groups at red and green bands were shown, but the unstressed values were found to have lower changes. The variance in the blue band values has a gentle slope with lesser changes than the other two bands. However, the blue band at V4 had similar changes like the other bands (Figure 6c).

The red variance had no differences between the unstressed and drought-treated on vertical angled images at V4 and horizontal images at V2, while the difference of green variance was shown at V2 and V3. Lastly, the blue variance was significantly different at every angle and vegetative stage (Table 4). They also showed more mixed positions than other image-based traits (Figure 7). The missing data of V3 before drought treatment, caused due to system errors, were excluded (Figure 7a).

### 2.5. t-SNE

The t-SNE plot was performed from three categorized image-based traits (Figure 8). It demonstrates the distance between the unstressed and drought-treated 13 image-based traits through dimension reductions. The traits before drought treatment and after recovery were close in distance, while those during drought treatment and recovery periods were relatively far. The largest difference was shown between the unstressed and drought-treated at the recovering period. This result was dissimilar to the area and boundary traits (Figure 3 and Figure 5).

## 3. Discussion

Quantified image-based traits enabled a more detailed and accurate evaluation of traits. However, each trait lacked a characteristic assessment, such as ignoring the area under the canopy to derive the results using single-angled image data. The area and boundary related traits were found to have significant differences between the unstressed and drought-treated individuals, while the classification among varieties was poor, or while it was possible to observe the differences among varieties, the determination of drought treatment was ambiguous, conversely (Table 2 and Table 3). It seems that a lack of some information in the dataset, such as canopy shape or growth habits, influenced the results.

The color-related traits, the mean color variance of the leaves, also varied in each variety. Traits for the estimation of drought stress, such as leaf area and biomass, may be evaluated using the area-related traits and boundary related traits of each angle. Color related traits comprise variations in mean color values. These traits are expected to be able to estimate the chlorophyll content of plants and nutrient status, as well as properties such as photosynthesis [21]. In Figure 6, the fluctuations in the green and red bands that comprise yellow colors vary throughout the drought stress treatment period [22].

Furthermore, it is evident that using individual image-based traits allows for a simpler and quicker process. The advantage of simplifying targeted traits is that it can streamline data production, enabling a faster analysis process. However, the same advantages and disadvantages apply when using a single angled image. One single image-based trait might have a lack of explanation for some trait evaluations. For instance, results using color data can only check the color of a plant’s organs, but the drying stress response of a plant is estimated by encompassing both physiological and morphological indicators which color values might be impossible to explain. Similarly, if only area or boundary data is used individually, plant status indicators such as leaf area and biomass may be assessed. However, it may be challenging to determine whether the observed phenomenon is due to a growth disorder or drought stress. Although an estimate can be made, distinguishing between the two causes can be difficult.

Therefore, we suggest a more precise estimation would be possible using indicators of different types together rather than using only one indicator. All image-based traits were used to simultaneously evaluate the drought resistance per variety (Figure 8). The t-distributed stochastic neighbor embedding (t-SNE) is one of the methods for dimensionality reduction. The principal component analysis (PCA), which is a linear analysis method, has some possibilities for unclear distributions between the clusters during dimensionality reduction. Therefore, we used t-SNE to avoid this issue. The visualized t-SNE plot demonstrated that these traits were significantly affected, and it may be inferred that there was a significant difference between the unstressed and the drought-treated group after recovery. Moreover, significant differences were observed between the unstressed and drought-treated during the recovery period in the t-SNE plots. The t-SNE plots resulted in the discriminable distances between the clusters from the start of drought treatment to the end of recovery, while some of the individual traits showed overlapping clustering zones even during the drought treatment (Figure 7). Thereby, we might suggest that a more accurate analysis on determining the drought stress will be possible by using image-based traits of various categories together.

We applied an image-based phenotyping system to optimize the selection of image-based soybean traits affected by drought in its early vegetative stages. It is possible to evaluate the characteristics of each variety and investigate the different developments in growth under drought and adequate conditions. Although the damage caused by drought during the early vegetative stages did not appear to be significantly different after the recovery period, we found that it was feasible to determine the degree of quantitative difference between individuals grown under adequate conditions and those that were drought-stressed by combining image-based traits.

Image-based phenotype analysis is considered to have sufficient potential for assessing characteristics related to drought stress response and varieties of soybean crops. The response to drought stress was different depending on the genotype and the timing of drought. Therefore, using one image-based trait was limited in evaluating the drought stress in the entire soybean population. We were able to assess drought damages using all acquired image traits, and the t-SNE algorithm identified aspects of recovery after drought treatment. The use of all image-based traits was able to discriminate the difference in recovery between individuals who visually found nearly equal levels of plant vigor after the recovery period. This evaluation of drought stress, without selecting a specific target trait by varieties, is expected to help reduce bottlenecks in the application of current high-throughput phenotyping methods and accelerate studies for precision agriculture with a satisfying efficiency.

## 4. Materials and Methods

### 4.1. Plant Materials and Experiments

Soybeans (*Glycine max* L.) were grown in an automated greenhouse (LemnaTec, Germany) of the Rural Development Administration (RDA, Korea) (Figure 9). Parents of 28 NAM populations of soybean, which were also provided by RDA (Table 5), were grown until the end of their growth cycle.

Due to the spatial limits of the conveyor, the experiment was divided into three sub-experiments, based on the vegetation stages (V2, V3, and V4) of the drought treatment (Table 6). A total of seven randomized plots of replications were created (two for unstressed, and five for drought-treated) comprising 28 plots of each variety. There was one individual plant in one pot of carrier, with a total of 28 individuals and 28 varieties per plot. However, 16 varieties of the second plot of unstressed plants were excluded from the experiment due to spatial limitations. Therefore, a total of 180 individuals were grown. A soil sensor was placed in one pot in each plot, under the same conditions as the other samples, at the center of each conveyor belt were plants were placed. This was done to monitor soil moisture and temperature.

The air temperature was maintained from 28 to 30 °C during the entire experiment, from 21 June 2019 to 3 March 2020, with a self-programmed automatic greenhouse control system. The greenhouse control system generalized also light, humidity, and ventilation. All the plants were removed from the conveyor belt after the final scheduled imaging, then raised until the end of their life cycle.

The number of nodes, the number of produced branches, and height were measured in each period as an actual measurement for validation and drought evaluation. The number of total nodes, main stem nodes, branches (data not used), and height were counted in 7-day intervals. In addition, the number of produced pods were measured at the end of their life cycle. These traits measured were conducted for the validation of drought treatment and image-based data.

The soybeans were watered evenly until the time of each vegetative stage in the 1st, 2nd, and 3rd subexperiments. A water deficit treatment of 14 days was carried out on the five treatment plots when 90% of the plant individuals were in the vegetation stages 2, 3, and 4, in which 14 days of 5% soil moisture was the maximum limit of permanent wilting point under our greenhouse environment. The unstressed plots, on the other hand, were fully watered. The soil moisture of the treatment plot was restricted to under 5%. The soil moisture was measured using an auto soil moisture meter WP-700C (MIRAE SEN-SOR, Korea). To prevent death caused by reaching the permanent wilting point, 15 mL of water was administered to the drought-treated plants that had reached severe dehydration levels, which were close to being fatal. All the plants were fully watered during the 14 days of recovering period.

### 4.2. Image Acquisition

In all three subexperiments, image data was acquired from the imaging chamber (Figure 9b) of the facility (LemnaTec, Aachen, Germany). The RGB images were taken at both vertical and horizontal angles. The image acquisition was conducted once a day each from before drought treatment to the end of drought treatment (Table 6). However, considering imaging data differed due to facility schedules, common date was acquired, which is necessary for analyzing the drought-affected phenotypes of the three subexperiments.

### 4.3. Image Processing and Analysis

The image analysis process comprises two large steps—preprocessing, and data processing. The preprocessing step includes the minimization of the error occurrences of quantified measures from the image analysis results. The study used an algorithm based on Python 3.8 [23] to crop images, remove other materials and their background, and set the image scales. Image crop is the first step of preprocessing which simultaneously resizes and standardizes the size of plant material images acquired from top/side angles for the stable operation of high-throughput data processing. An object area selection procedure was then carried out for separating the targeted plant area from obstacles in the image, such as pot, carrier, and background. The scale set was finally processed to set the uniform ratio of pixels to be the same value in all images and to prevent errors of distortion derived by sensors.

The quantified morphological parameters were then generated from primarily processed plant images. The traits extracted from the images were classified into three categories: area, boundary, and color (Table 7). This classification was according to each creation process, and what they indicate.

Most of the traits in ‘Area’ indicate the number of pixels in the projected object. However, as an exception, the ‘caliper length’ represents the distance between the longest end points on the projected area. It was assessed to be closer to the ‘Area’ category than the ‘Boundary’ from its calculation process (Figure 10a).

In the case of ‘Boundary’ category, it indicates the perimeter, the ratio of the perimeter to the area, and the min enclosing circle. These traits were secondarily calculated from the primal parameters such as area. In addition, in the case of ‘min enclosing circle diameter’, which was derived from the projected area of the plant, it was included in ‘Boundary’ depending on which is more related (Figure 10b).

The ‘Color’ traits indicate pixel values in the obtained image. RGB image can be separated into three images of red, green, and blue bands. The pixels are valued from 0 to 255, which total 256 levels. These values refer to the color and brightness of one pixel in the image. However, since the captured image included a support for fixing the plant, using each pixel value itself to analyze the process had some issues on time and efficiency in the analysis process. Therefore, rather than using only the pixel values of each band, the variance of each band was used in the analysis.

The production of image-based traits was achieved through the image processing software Image J (Ver. 1.52a, National Institutes of Health, Bethesda, MD, USA) [24]. The image analysis process begins with noise removal; a process for preventing incomplete elimination that might occur in preprocessing. The program then separates the images into red, green, and blue channels to create binary images of each band and develops their color distribution data. Produced binary images then merge into one grayscale band, resulting in the development of the region of interest (ROIs) of the targeted object in the image through the Otsu color threshold [25]. Shape measurement is then conducted by overlaying the created ROIs to RGB images and the quantified parameters are produced through the calculation of the measured values. The image analysis process ends with saving the quantified shape and color data into .csv format files through macros of Image J.

### 4.4. Statistical Analysis

The quantified image data were analyzed using the program R (Ver. 4.2.2, R Core Team, 2021). Unnecessary data with too few measurements or too many errors were excluded from the analysis. The outliers were removed in both image and hand-measured datasets using quantile regressions.

The Kruskal–Wallis analysis, PCA, and t-SNE algorithm were conducted from R programming. Missing data and outliers were removed from the datasets. We used all datasets to perform t-SNE algorithm, and the embedding dimension was set as 2, and perplexity as 30. Then, t-SNE algorithms were visualized, which enabled the detection of comprehensive differences between the unstressed and drought-treated samples to determine the most affected vegetative stages and angles for precise image analysis [26].

## 5. Conclusions

Increasing the production of key crops that contribute to food security is taking a toll on climate change due to global warming. The most efficient way to solve this problem is through breeding resistant varieties. Gene editing technology and high-throughput phenotyping have emerged to supplement and replace the limitations of existing breeding methods. However, high-throughput phenotype analysis currently suffers from bottlenecks in application depending on the crop.

Therefore, we focused on developing an efficient process for stress evaluation from drought-stressed soybeans. Phenotypes and recovery rates related to drought stress were evaluated by an image-based phenotype analysis platform. We were able to evaluate drought stress on a soybean population, regardless of the diversity of genotypes. The analysis, using the phenotypes measured in the images, enabled an evaluation that was not significantly affected by differences in breeds. This might suggest a plan to solve the difficulties in applying high-throughput phenotyping methods due to the variation in traits depending on their varieties and species.

## Figures and Tables

**Figure 1 plants-12-02331-f001:**
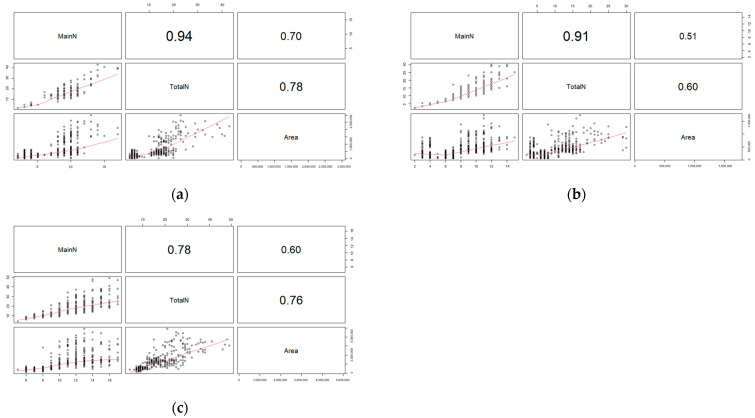
The Pearson correlation coefficient analysis plot for number of main stem nodes, total nodes, and vertical image-based area. Red line is the linear correlation line. (**a**) is measured at 2nd vegetative stage; (**b**) is measured at 3rd vegetative stage; and (**c**) is measured at 4th vegetative stage.

**Figure 2 plants-12-02331-f002:**
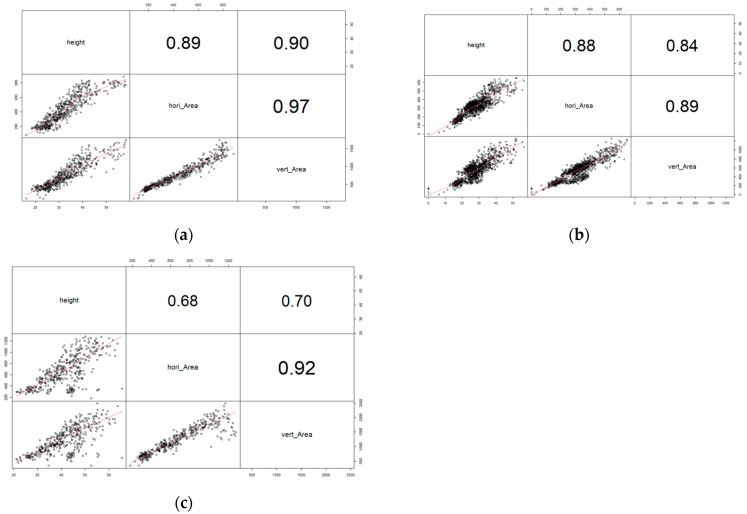
The Pearson correlation coefficient analysis plot for height, horizontal image-based area, and vertical image-based area. Red line is the linear correlation line. (**a**) is measured at 2nd vegetative stage; (**b**) is measured at 3rd vegetative stage; and (**c**) is measured at 4th vegetative stage.

**Figure 3 plants-12-02331-f003:**
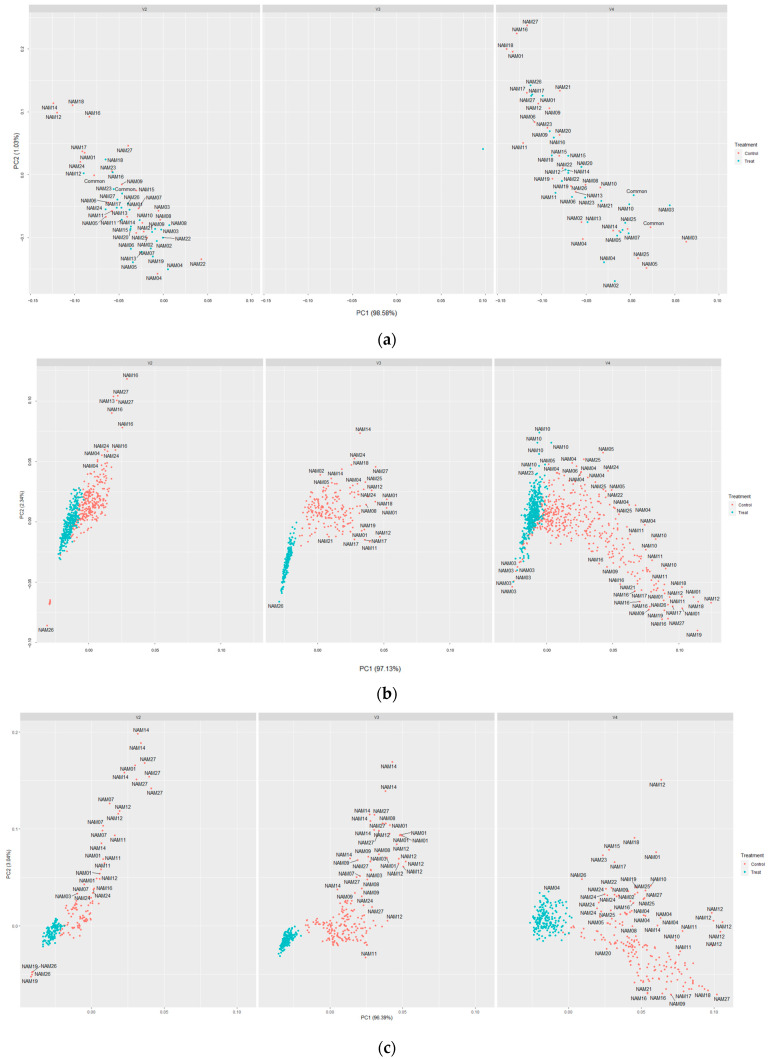
Principal component analysis plot for area related traits in vertical angle images. The red colored points are the unstressed and the blue colored points are drought treated. Left is the V2, middle is the V3, and right is the V4. (**a**) Before drought treatment; (**b**) after drought treatment; (**c**) recovering period (7 days after recovery started); and (**d**) after the full recovery (14 days) was done.

**Figure 4 plants-12-02331-f004:**
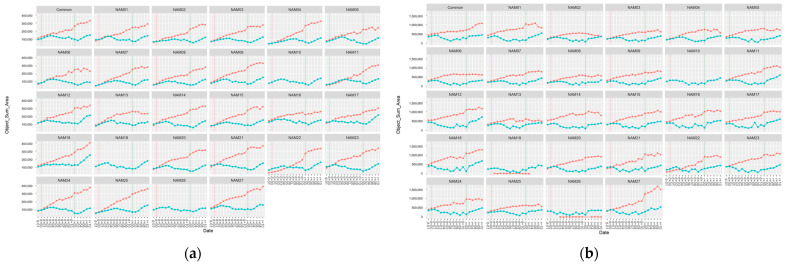
The ‘object sum area’ from vertical and horizontal angle images at V2. The red colored lines are the unstressed and the blue colored lines are the drought treated. (**a**) The ‘object sum area’ from V2 horizontal angled image; (**b**) the ‘object sum area’ from V2 vertical angled image.

**Figure 5 plants-12-02331-f005:**
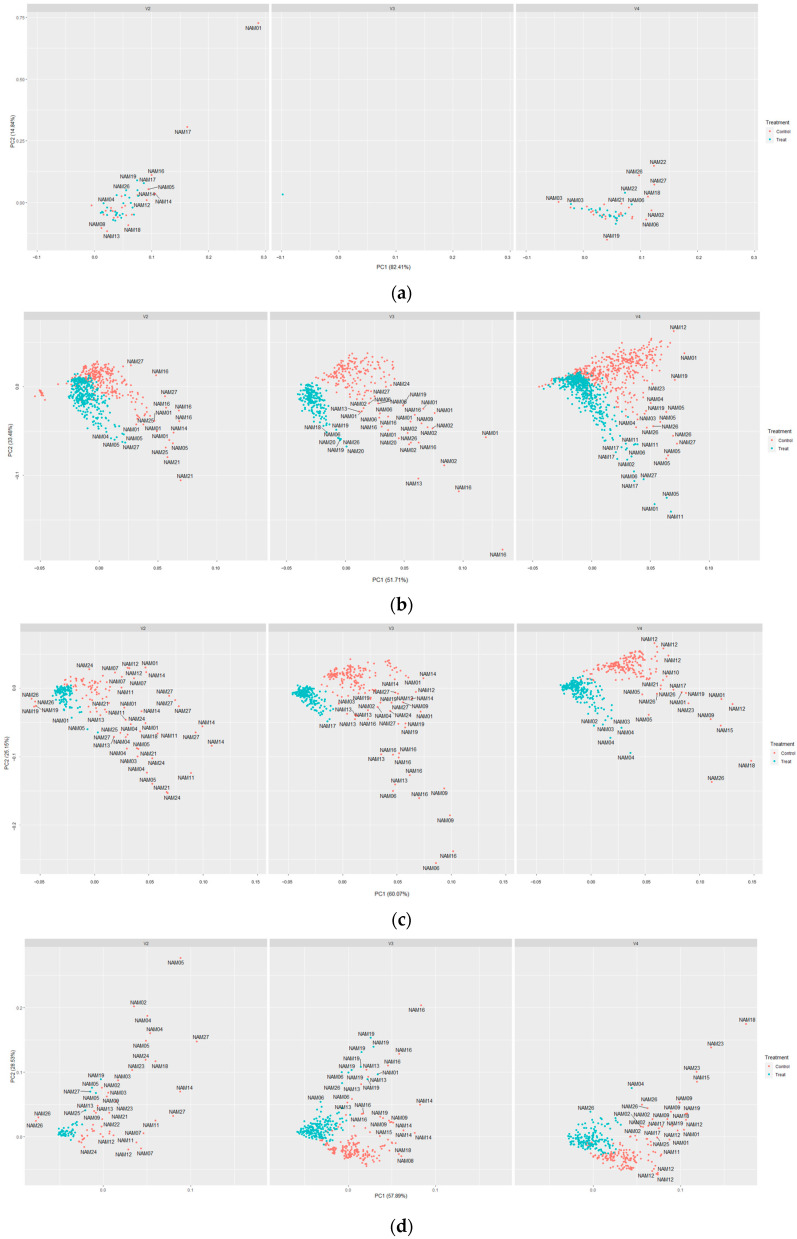
Principal component analysis plot for boundary related traits. The red colored points are the unstressed and the blue colored points are drought treated. Left is the V2, middle is the V3, and right is the V4. (**a**) Before drought treatment; (**b**) after drought treatment; (**c**) recovering period (7 days after recovery started); and (**d**) after the full recovery (14 days) was done.

**Figure 6 plants-12-02331-f006:**
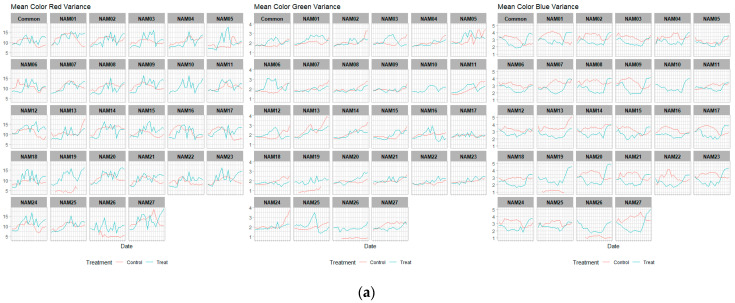
Line plot for color variances at all vegetative stage sets. The red colored lines are the unstressed and the blue colored lines are drought treated. Left is the red variance, middle is the green variance, and right is the blue variance. (**a**) 2nd vegetative stage (V2); (**b**) 3rd vegetative stage (V3); (**c**) 4th vegetative stage (V4).

**Figure 7 plants-12-02331-f007:**
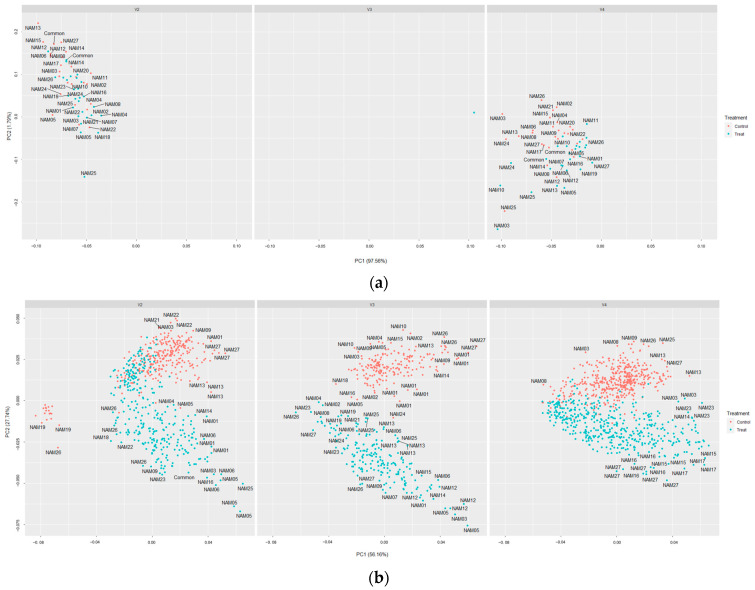
Principal component analysis plot for color related traits. The red colored points are the unstressed and the blue colored points are drought treated. Left is the V2, middle is the V3, and right is the V4. (**a**) Before drought treatment; (**b**) after drought treatment; (**c**) recovering period (7 days after recovery started); and (**d**) after the full recovery (14 days) was done.

**Figure 8 plants-12-02331-f008:**
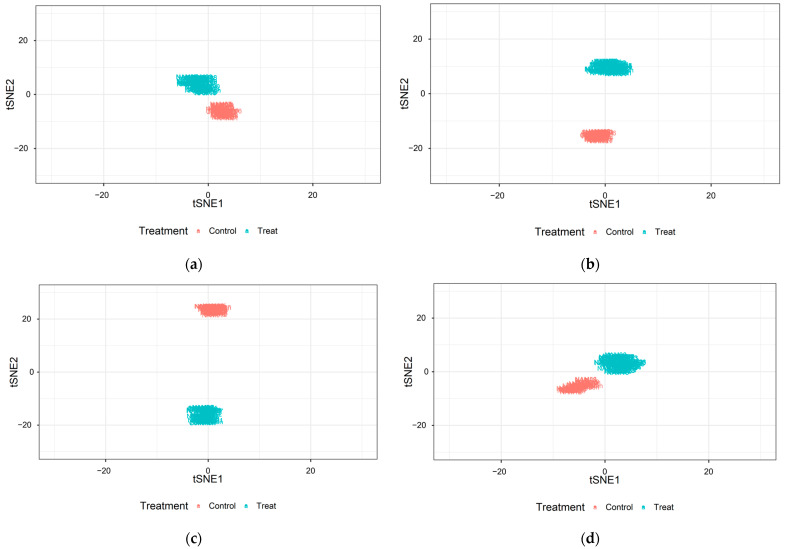
The plots performed from t-distributed stochastic neighbor embedding (t-SNE) algorithms using 13 image-based traits at 3rd vegetative stage. The red colored points are the unstressed and the blue colored points are drought-treated. (**a**) before drought treatment; (**b**) after drought treatment; (**c**) recovering period (7 days after recovery started); and (**d**) after the full recovery (14 days) was done.

**Figure 9 plants-12-02331-f009:**
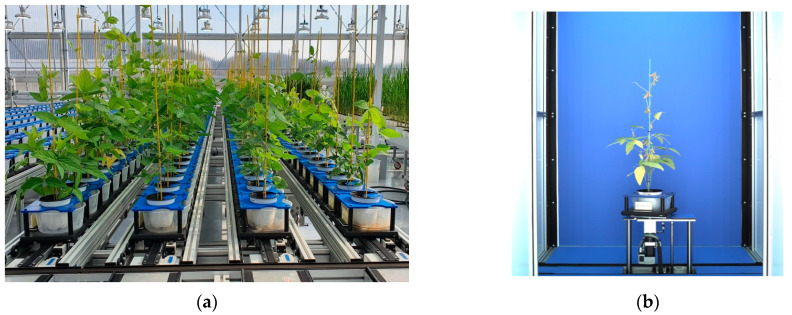
The automated greenhouse phenotyping system in Rural Development Administration (RDA, Korea) and image acquisition process in the imaging chamber. (**a**) The automated greenhouse with a conveyor belt; (**b**) the inside view of the imaging chamber for phenotyping.

**Figure 10 plants-12-02331-f010:**
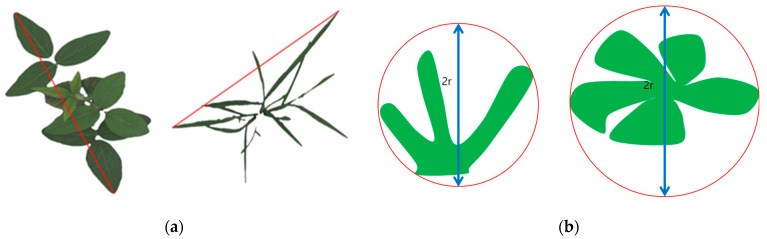
The visualized scheme of ‘Caliper length’ and ‘Min enclosing circle diameter’. (**a**) is the ‘Caliper length’ derived from projected area; (**b**) is the ‘Min enclosing circle diameter’ derived from the min enclosing circle.

**Table 1 plants-12-02331-t001:** Kruskal–Wallis analysis of variance for validating drought influences by height.

Days after Drought Treatment (Days)		Stages	*p* Value (<0.05) *^,^ **			
			Height	Main N	Total N	Pods
Before drought treatment	(−7)	V2	NA	NA	NA	NA
		V3	0.5644	NA	NA	NA
		V4	0.2909	NA	NA	NA
1st day of drought treatment	(0)	V2	NA	0.4895	0.5968	NA
		V3	0.3319	NA	NA	NA
		V4	0.6659	NA	NA	NA
During drought treatment	(+7)	V2	<0.05	<0.05	<0.05	NA
		V3	<0.05	<0.05	<0.05	NA
		V4	0.2513	<0.05	<0.05	NA
End of drought treatment/Recoverying started	(+14)	V2	<0.05	<0.05	<0.05	NA
		V3	<0.05	<0.05	<0.05	NA
		V4	<0.05	<0.05	<0.05	NA
During recovery	(+21)	V2	<0.05	<0.05	<0.05	NA
		V3	<0.05	<0.05	<0.05	NA
		V4	<0.05	0.0522	<0.05	NA
End of recoverying period	(+28)	V2	<0.05	0.3395	0.8427	0.01479
		V3	<0.05	0.1934	0.4445	0.7254
		V4	<0.05	<0.05	<0.05	0.6582

* The two NAs at V2 stage, in height, were validated from counting main stem, total nodes, or area. ** Statistically different at the <0.05 significance level, Kruskal–Wallis analysis of variance.

**Table 2 plants-12-02331-t002:** The Kruskal–Wallis analysis of area image-based traits of each vegetative stage for the 28 NAM population parent varieties.

Image-Based Traits	Angle	Vegetative Stage	*p* (<0.005)
Area	Top	V2	<0.005 *
		V3	0.006714
		V4	<0.005 *
	Side	V2	<0.005 *
		V3	0.006255
		V4	<0.005 *
Caliper length	Top	V2	0.007195
		V3	<0.005 *
		V4	<0.005 *
	Side	V2	<0.005 *
		V3	<0.005 *
		V4	<0.005 *
Convex hull area	Top	V2	<0.005 *
		V3	0.009479
		V4	<0.005 *
	Side	V2	<0.005 *
		V3	0.0006882
		V4	<0.005 *
Min area rectangle area	Top	V2	<0.005 *
		V3	0.007009
		V4	<0.005 *
	Side	V2	<0.005 *
		V3	<0.005 *
		V4	0.00161
Object sum area	Top	V2	<0.005 *
		V3	0.00297
		V4	<0.005 *
	Side	V2	<0.005 *
		V3	0.006255
		V4	<0.005 *

* Statistically different at the <0.005 significance level, Kruskal–Wallis analysis of variance.

**Table 3 plants-12-02331-t003:** The Kruskal–Wallis analysis of boundary image-based traits of each vegetative stage for the 28 NAM population parent varieties and drought treatment.

Image-Based Traits	Variables	Angle	Vegetative Stage	*p* (<0.005)
Boundary point roundness	Drought treatment	Top	V2	0.4897
			V3	0.02647
			V4	0.3807
		Side	V2	<0.005 *
			V3	<0.005 *
			V4	<0.005 *
Circumference	Varieties	Top	V2	<0.005 *
			V3	<0.005 *
			V4	<0.005 *
		Side	V2	<0.005 *
			V3	<0.005 *
			V4	<0.005 *
Convex hull circumference	Varieties	Top	V2	<0.005 *
			V3	0.008018
			V4	<0.005 *
		Side	V2	<0.005 *
			V3	<0.005 *
			V4	<0.005 *
Min enclosing circle diameter	Varieties	Top	V2	<0.005 *
			V3	0.005517
			V4	<0.005 *
		Side	V2	<0.005 *
			V3	<0.005 *
			V4	<0.005 *
Roundness	Varieties	Top	V2	0.4303
			V3	0.001177
			V4	<0.005 *
		Side	V2	0.007898
			V3	<0.005 *
			V4	<0.005 *

* Statistically different at the <0.005 significance level, Kruskal–Wallis analysis of variance.

**Table 4 plants-12-02331-t004:** The Kruskal–Wallis analysis of color image-based traits of each vegetative stage for the drought treatment.

Image-Based Traits	Angle	Vegetative Stage	*p* (<0.005)
Mean color red variance	Top	V2	<0.005 *
		V3	<0.005 *
		V4	0.1753
	Side	V2	0.2799
		V3	<0.005 *
		V4	<0.005 *
Mean color green variance	Top	V2	0.1009
		V3	<0.005 *
		V4	<0.005 *
	Side	V2	<0.005 *
		V3	0.308
		V4	<0.005 *
Mean color blue variance	Top	V2	<0.005 *
		V3	<0.005 *
		V4	<0.005 *
	Side	V2	<0.005 *
		V3	<0.005 *
		V4	<0.005 *

* Statistically different at the < 0.005 significance level; Kruskal–Wallis analysis of variance.

**Table 5 plants-12-02331-t005:** The soybean varieties of NAM population parents.

Numbering	Variety
Common	Daepung
NAM 01	Bangsa
NAM 02	Pungwon
NAM 03	Hannam
NAM 04	Sowon
NAM 05	Galche
NAM 06	Somyeong
NAM 07	Sinhwa
NAM 08	Pureun
NAM 09	Taegwang
NAM 10	Wuram
NAM 11	Danbek
NAM 12	PI96983
NAM 13	Haman
NAM 14	Willians82
NAM 15	Saedanbek
NAM 16	Daewon
NAM 17	Hwanggeum
NAM 18	Chungja
NAM 19	Chungja 3ho
NAM 20	Sochung 2ho
NAM 21	Ilpumgeomjung
NAM 22	Daeheuk
NAM 23	Josangseori
NAM 24	Yeunpung
NAM 25	Chunal
NAM 26	Heukchung
NAM 27	Seoritae

**Table 6 plants-12-02331-t006:** The table of experimental schedule.

Process *	Features	Materials
Data acquisition	V2 drought treatment V3 drought treatment V4 drought treatment	Imaging chamber
Preprocessing	Image crop Object area selection Scale settings	Python
Data processing	Noise removal Channel separation Binary image creation Region of Interest Shape measurement Color measurement	Image J
Data analysis	Outlier detection and removal Data validation Data analysis	R programming

* The process follows the order that starts from data acquisition to data analysis.

**Table 7 plants-12-02331-t007:** Estimated image-based traits from plant images produced by ImageJ.

Types	Image-Based Traits	Features
Area	Area	Number of pixels in projected area.
	Caliper Length	The longest distance in the object.
	Convex Hull Area	Number of pixels in convex hull area.
	Min Area Rectangle Area	Number of pixels in an area of the smallest rectangle that can contain the projected object.
	Object Sum Area	The sum of the numbers of the pixel of all projected objects in the image.
Boundary	Boundary Point Roundness	The ratio of the boundary points of the object to the area of the circle that diameter is equal to the maximum diameter of the object.
	Circumference	Circumference of the smallest circle that can contain the projected object.
	Convex Hull Circumference	Circumference of the smallest circle that can contain a convex hull.
	Min Enclosing Circle Diameter	Diameter of the smallest circle that can contain the projected object.
	Roundness	The ratio of the object to the area of the circle that diameter is equal to the maximum diameter of the object.
Color	Mean Color Red Variance	Variance in mean R values in projected object.
	Mean Color Green Variance	Variance in mean G values in projected object.
	Mean Color Blue Variance	Variance in mean B values in projected object.

## Data Availability

Data available on request due to restrictions.

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
