# Peer review of "Comparison of Various Drought Resistance Traits in Soybean (Glycine max L.) Based on Image Analysis for Precision Agriculture"

_plants, 2023, doi:10.3390/plants12122331_

Round 1
Reviewer 1 Report
1. The relationship between the three trait categories is not shown. Can you explain the relationship between the various image-based traits?
2. To explain traits for which data is missing due to system errors, is there any additional data from before the drought treatment at V3?
3. RGB values, especially the green and red values, are known as powerful indicators for plant health and stress evaluation. Is there a reason why only color variances are used, while RGB color values are excluded?
4. The color variance is an important factor for drought evaluation. However, their features were briefly explained in one paragraph during the discussion. Is there additional explanation for color variables?
5. What are the things that can be explained when each image-based trait is used as a single, and what are the pros and cons of each?
6. What are the differences between the area and boundary related traits? They both seem to be in the "area" category. Please add more explanations for the differences between these trait categories.
7. Only image-based traits are informed in this manuscript. For validation, it might need comparisons with some actual measured soybean traits. Is there any information about yield and other drought related traits by actual measurement?
Author Response
Reviewer 1.
- The relationship between the three trait categories is not shown. Can you explain the relationship between the various image-based traits?
> Done as the reviewer suggested. We’ve added more explanations in Materials & methods section. Thanks for your kind advice.
- To explain traits for which data is missing due to system errors, is there any additional data from before the drought treatment at V3?
> The vegetation traits such as the number of nodes and height are measured before drought treatment, and soil moisture was measured from the day after seeding to end of their drought treatment. However, vegetation traits data such as node and height were not added because they were not used in the current paper.
- RGB values, especially the green and red values, are known as powerful indicators for plant health and stress evaluation. Is there a reason why only color variances are used, while RGB color values are excluded?
> Analyzing the RGB values is certainly a strong indicator of drought. However, the evaluation of RGB values, requires a large amount of image files to be analyzed one by one. The processing all images enlarges the time and hardware requirements, therefore, we used the mean variance of RGB values is used for quick and efficient evaluation process. Thanks for your kind advice.
- The color variance is an important factor for drought evaluation. However, their features were briefly explained in one paragraph during the discussion. Is there additional explanation for color variables?
> Done as the reviewer suggested. Thanks for your kind advice.
- What are the things that can be explained when each image-based trait is used as a single, and what are the pros and cons of each?
> Clearly, there are advantages on using individual image-based traits. By simplifying targeted traits, the compressed data production may allow accelerated analyzing process.
However, to explain each image-based trait, some traits cannot be explained in one image-based trait can be explained. For instance, results using color data can only check the color of a plant organs, but the drying stress response of a plant is estimated by encompassing both physiological and morphological indicators which color values might be impossible to explain. Conversely, if only area or boundary data is used individually, plant status such as leaf area and biomass can be estimated, but difficult to determine the phenomenon is due to whether the growth disorder or drought stress.
Therefore, we suggest the more precise estimation would be possible by using indicators of different types together rather than using only one indicator. Thank you for the kind advice.
- What are the differences between the area and boundary related traits? They both seem to be in the "area" category. Please add more explanations for the differences between these trait categories.
> Done as the reviewer suggested. We’ve added more explanations in Materials & methods section. Thanks for your kind advice.
- Only image-based traits are informed in this manuscript. For validation, it might need comparisons with some actual measured soybean traits. Is there any information about yield and other drought related traits by actual measurement?
> There is actual data were measured at 7-day intervals. The number of main stem nodes, the total number of nodes, the number of branches, and height were measured. Additionally, the number of produced pods were counted at the end of their lifecycles to evaluate drought effects and further studies. The actual measured data were used as a validation of the data, and mention for those measurements have been added in the materials & methods section as you’ve suggested. Thank you for the kind advice.
Reviewer 2 Report
The article titled "Comparison of various drought resistance traits in soybean (Glycine max L.) based on image analysis for precision agriculture" presents and evaluates, using image-based phenotypes, methods for monitoring tolerant trait, drought stress assessment and recovery rate, considering different early vegetative stages of soybean. In this paper, the authors used principal component analysis and t-distributed stochastic neighbor embedding algorithms.
The work describes the methodology in full and correct form. The results are displayed in very clear tables and graphs. The objective and the development of the work are both relevant to the scientific community and of interest in very different areas, mainly for rapid climate changes and global warming currently.
The article titled "Comparison of various drought resistance traits in soybean (Glycine max L.) based on image analysis for precision agriculture" presents and evaluates, using image-based phenotypes, methods for monitoring tolerant trait, drought stress assessment and recovery rate, considering different early vegetative stages of soybean. In this paper, the authors used principal component analysis and t-distributed stochastic neighbor embedding algorithms.
The work describes the methodology in full and correct form. The results are displayed in very clear tables and graphs. The objective and the development of the work are both relevant to the scientific community and of interest in very different areas, mainly for rapid climate changes and global warming currently.
Author Response
The article titled "Comparison of various drought resistance traits in soybean (Glycine max L.) based on image analysis for precision agriculture" presents and evaluates, using image-based phenotypes, methods for monitoring tolerant trait, drought stress assessment and recovery rate, considering different early vegetative stages of soybean. In this paper, the authors used principal component analysis and t-distributed stochastic neighbor embedding algorithms.
The work describes the methodology in full and correct form. The results are displayed in very clear tables and graphs. The objective and the development of the work are both relevant to the scientific community and of interest in very different areas, mainly for rapid climate changes and global warming currently.
> We've added explanations in the manuscript for image-based traits and results as you've suggested. Thank you for the kind advice.
Reviewer 3 Report
In sections 2.1, it seems unclear from figure 1 to distinguish the experiments in horizontal or vertical image traits. So how to conclude that the horizontal image traits were not separated by drought treatment?
Figure 5 shows the traits after recovery were close in distance. In sections 3, it is mentioned that the use of all image-based traits was able to discriminate the difference between individuals after the recovery period. Is there any more specific analysis for this?
Author Response
- In sections 2.1, it seems unclear from figure 1 to distinguish the experiments in horizontal or vertical image traits. So how to conclude that the horizontal image traits were not separated by drought treatment?
> The result from the horizontal image showed a relatively large overlapping cluster area, so it was excluded by insufficiency for explain the analysis using area traits. However, as you recommended, we also thought that it was not sufficient to explain the conclusions about their separation. So Figure 2, a raw data plot of the 'object sum area', one of the area traits, was newly added. In the horizontal image shown in the figure 2-a, the tendency of no separation between the two treatment groups appeared longer than the vertical image at the beginning of drought treatment.
Additionally, in the sentence [However, the horizontal area traits during drought treatment were not distinctly separated between unstressed and drought treated (Figure 1-b).], a typo in which vertical is written as horizontal has been corrected. Thank you for the kind advice.
- Figure 5 shows the traits after recovery were close in distance. In sections 3, it is mentioned that the use of all image-based traits was able to discriminate the difference between individuals after the recovery period. Is there any more specific analysis for this?
> The PCA and t-SNE using individual categories traits were performed in additional analyzes. However, their results were excluded because they were also thought to be explained by the formation of clusters of overlapping regions as in Figure 4, and issue for having too many figures in the manuscript. However, as reviewer 3 suggested, the explanation and comparison of t-SNE results were insufficient. We have therefore added that reference to Section 3 of this text. Thanks for your kind advice.